Word embedding for social sciences: an interdisciplinary survey

Matsui Akira 1 matsui-akira-zr@ynu.ac.jp
http://orcid.org/0000-0002-1942-2831 Ferrara Emilio 2
1 College of Business Administration, Yokohama National University , Yokohama, Kanagawa , Japan
2 Thomas Lord Department of Computer Science, University of Southern California , Los Angeles, California , United States
Alatas Bilal
Electronic publication date: 2024 Dec 5
Publication date: 2024
Volume: 10
Electronic Location ID: e2562
Received 2023 Nov 24; Accepted 2024 Nov 8
Copyright: © 2024 Matsui and Ferrara
Copyright year: 2024
Copyright holder: Matsui and Ferrara
License: This is an open access article distributed under the terms of the Creative Commons Attribution License, which permits unrestricted use, distribution, reproduction and adaptation in any medium and for any purpose provided that it is properly attributed. For attribution, the original author(s), title, publication source (PeerJ Computer Science) and either DOI or URL of the article must be cited.
License URL: https://creativecommons.org/licenses/by/4.0/

Keywords: Word embedding, word2vec, Computational social science, Bias in machine learning

Funding: JSPS KAKENHI Grant-in-Aid for Scientific Research JP22K20159 Research Institute of Science and Technology for Society, Japan JPMJRS23L4 This work was supported by JSPS KAKENHI Grant-in-Aid for Scientific Research (No. JP22K20159); Research Institute of Science and Technology for Society, Japan, Grant Number JPMJRS23L4. The funders had no role in study design, data collection and analysis, decision to publish, or preparation of the manuscript.

==============================
Machine learning models learn low-dimensional representations from complex high-dimensional data. Not only computer science but also social science has benefited from the advancement of these powerful tools. Within such tools, word embedding is one of the most popular methods in the literature. However, we have no particular documentation of this emerging trend because this trend overlaps different social science fields. To well compile this fragmented knowledge, we survey recent studies that apply word embedding models to human behavior mining. Our taxonomy built on the surveyed article provides a concise but comprehensive overview of this emerging trend of intersection between computer science and social science and guides scholars who are going to navigate the use of word embedding algorithms in their voyage of social science research.

Introduction

The advancement of machine learning technologies has revolutionized traditional research methodologies in the social sciences, enabling the extraction of insights from unconventional data sources1 . Recently, we have been witnessing the expansion of social science research that explores textual data, captivated by the concept of “Text as Data” (Gentzkow, Kelly & Taddy, 2019; Benoit, 2020; Grimmer & Stewart, 2013). This trend is evident not only in fields that traditionally rely on quantitative analysis, such as economics! (Cong, Liang & Zhang, 2018) and finance (Rahimikia, Zohren & Poon, 2021), but also in disciplines with a penchant for qualitative analysis, like history (Wevers & Koolen, 2020). The flourishing body of literature in this domain enables social scientists to meticulously select or develop machine learning models for text mining that meet the demands of their research, providing a vast array of options. While rule-based models, such as dictionary-based models, remain popular (Tausczik & Pennebaker, 2010; Hutto & Gilbert, 2014), neural network-based embedding models are gaining increasing traction in social science research.

Embedding models learn low-dimensional representations of entities from data by predicting relationships between these entities (Boykis, 2024). The advantage of embedding models in social science research lies in their ability to acquire the “embedded” structure in an unsupervised manner. This inherent flexibility allows social scientists to apply these models to their data with minimal constraints, facilitating their analyses. These entities could be nodes in a graph (Goyal & Ferrara, 2018; Barros et al., 2021; Cai, Zheng & Chang, 2018), pixels in an image (Liu et al., 2020; Baltrušaitis, Ahuja & Morency, 2018), or vertices in a network (Cui et al., 2018; Zhou et al., 2022a). Despite the multitude of embedding algorithms available for diverse purposes, word embedding models have been recognized as some of the most widely adopted methods (Chaubard et al., 2019; Mikolov et al., 2013a, 2013b; Goldberg & Levy, 2014; Mikolov, Yih & Zweig, 2013; Mikolov et al., 2015). Among these, the most accessible word embedding model for social science research is the word2vec model (Mikolov et al., 2013b; Mikolov, Yih & Zweig, 2013; Mikolov et al., 2013a), used in diverse fields ranging from management science (Lee & Kim, 2021; Toubia, Berger & Eliashberg, 2021; Chanen, 2016; Li et al., 2021; Bhatia et al., 2021; Chen & Zhang, 2023; Zheng, 2020; Khaouja et al., 2019; Tshitoyan et al., 2019) to poetry (Nelson, 2021).

The rise in the use of word2vec and other related models in social science research has not been accompanied by adequate documentation of how these models are utilized. However, because this trend spans multiple disciplines, we are witnessing a lack of sufficient cross-communication. First, this absence of an integrated knowledge framework hampers discussions about the broader impact and relevance of word embedding in social science research. Second, the absence of comprehensive surveys spanning multiple disciplines has led social scientists to reference only literature within their own fields, inadvertently leading to the reinvention of methodologies already developed in other domains. Consequently, this solid approach often results in deserving scholarly works not receiving the recognition and citation they merit, thus undermining their rightful academic evaluation. Therefore, we need to review the broad spectrum of studies that should be referenced, many of which fall outside their immediate expertise.

To address the knowledge gap resulting from a lack of connections between disciplines, this article aims to identify emerging interdisciplinary trends in machine learning models arising across diverse fields. In this literature review, we seek to bridge the knowledge gaps between different fields and ask the reseach question “How similarly are word embedding models, such as word2vec, applied across different methodologies in social science research?” This literature review builds a taxonomy of these application methodologies to illustrate how social scientists harness word embedding models, particularly focusing on word2vec. The taxonomy emphasizes the applications of word embedding published in social science journals, outlining the contemporary landscape of social science research utilizing word embedding models, rather than delving into the technical intricacies of these neural network models. This literature review targets social scientists who are currently using or considering the use of word embedding models like word2vec in their research. Therefore, the presented article does not comprehensively review the state of the art in the technical aspects of embedding models. Instead, we offer a review that covers various social science disciplines, especially for those who may not specialize in machine learning or natural language processing. To provide an overview of such different fields, we develop a taxonomy to guide potential readers in situating their studies within this interdisciplinary literature.

This interdisciplinary survey covers literature published across a wide range of research fields, helping researchers or practitioners access studies and insights outside their own areas of expertise. In addition, the taxonomy developed in this survey helps researchers in positioning their work within the context of the literature, which is too broad to be referred in a article in a single field. In this way, the interdisciplinary literature review offered here contributes to bridge barriers among research fields. This survey also helps us understand and enhance the application of word embedding models in social science contexts.

The structure of this survey is as follows. After the discussion of search methodology, we begin this survey with a brief overview of the technical background of word2vec in “Word Embedding Models”. Then, we introduce the nine labels used in our taxonomy (Table 1) and review the articles detailing the taxonomy throughout “Pre-trained Models” to “Non-text”. We discuss a pitfall of similarity measurements with a simple experiment. In addition, we employ the proposed taxonomy to understand the literature using the word2vec model for social science.

Table 1 Labels for analysis methods with word embedding.

Labels	Definition	
Pre-trained	Pre-trained model	
Working variable	Variables constructed from a word embedding model for analysis	
W/theory	Validating a theory proposed in a research field for analysis, or quoting it to justify an analysis.	
W/reference	Using references such as the words with specific semantics to define	
Non-text	Applying word embedding model to non-text data (e.g., human behavior data)	
Same words	Comparing the same words from different models	
Human subj	Employing human subjects for analysis	
Clustering	Clustering analysis using word embedding vectors	
Prediction	Prediction using word embedding vectors	

Survey methodology

This survey primarily focuses on articles published in social science journals, aiming to conduct a concentrated literature survey. Although our main attention was not directed toward research in word embedding published in computer science conference proceedings, we selectively included some of these articles to ensure a comprehensive understanding of the field. This approach was essential to paint a complete picture of the literature landscape.

We primarily focused on articles that had a clear application or discussion of word embedding models in a social science context. We gave preference to articles published in peer-reviewed journals and conference proceedings to guarantee the credibility and academic rigor of our sources. Conversely, we excluded articles that did not fall within our specified subject areas. Non-English publications were also excluded due to language constraints. This exclusion was necessary to ensure that our sources were reproducible and verifiable by our readers.

For this survey, we predominantly used the Scopus Database, implementing a specific query that included terms word2vec and combinations such as word AND embedding. To ensure that our survey was as exhaustive as possible, we expanded our search to include Google Scholar and the Social Science Research Network (SSRN) to include working articles or preprints, adapting our query to suit their unique search algorithms. This process is not limited to research articles but also includes other types of publications, such as articles, book chapters, and review articles. Additionally, we compile articles that cite (Mikolov et al., 2013b; Mikolov, Yih & Zweig, 2013) to capture the literature on the application of word2vec. From the collected articles, we select those published in fields related to social sciences. The specific fields we cover are listed in the research area column in Tables 2 and 3.

Table 2 Summary of the analytical methods with word embedding for the social science research I.

	Research area	Pre-trained	Define variable	W/theory	W/reference	Non text	Same words	Human subj	Clustering	Prediction	
Chersoni et al. (2021)	Neuro. Sci	✓		✓							
Kelly et al. (2020)	Cogni. Sci			✓							
Caliskan, Bryson & Narayanan (2017)	Cogni. Sci		✓		✓			✓			
Kroon, Trilling & Raats (2021)	Communication		✓		✓						
Kroon et al. (2020)	Communication		✓		✓						
Bhatia & Bhatia (2021)	Communication				✓						
Wang (2019)	Finance									✓	
Jha, Liu & Manela, 2021	Finance										
Lee & Kim (2021)	Humanity									✓	
Čech et al. (2019)	Linguistic		✓	✓							
Toubia, Berger & Eliashberg (2021)	Manag. Sci		✓				✓				
Chanen (2016)	Manag. Sci						✓				
Li et al. (2021)	Manag. Sci				✓						
Bhatia et al. (2021)	Manag. Sci	✓						✓		✓	
Chen & Zhang (2023)	Manag. Sci									✓	
Zheng (2020)	Manag. Sci	✓							✓		
Khaouja et al. (2019)	Manag. Sci								✓		
Tshitoyan et al. (2019)	Mat. Sci		✓			✓				✓	
Nelson (2021)	Cultural Studies		✓		✓						
Nanni & Fallin (2021)	Cultural Studies								✓		
Rheault & Cochrane (2020)	Poli. Sci		✓			✓					
Gennaro & Ash (2022)	Poli. Sci		✓		✓						
Rodman (2020)	Poli. Sci				✓						
Rozado & al Gharbi (2021)	Poli. Sci		✓		✓						
Leavy, Keane & Pine (2019)	Poli. Sci										
Ash, Chen & Ornaghi (2020)	Poli. Sci		✓								
Nyarko & Sanga (2020)	Poli. Sci		✓								

Table 3 Summary of the analytical methods with word embedding for the social science research II.

	Research Area	Pre-trained	Define variable	W/theory	W/reference	Non text	Same words	Human subj	Clustering	Prediction	
Kim & Shin (2017)	Psychology		✓		✓						
Peterson, Chen & Griffiths (2020)	Psychology			✓							
Boyce-Jacino & DeDeo (2020)	Psychology		✓								
Richie et al. (2019)	Psychology	✓						✓		✓	
Soni, Lerman & Eisenstein (2021)	Sci. metrics						✓				
Hu et al. (2018)	Sci. metrics		✓								
Kozlowski, Taddy & Evans (2019)	Sociology	✓	✓		✓						
Jones et al. (2020)	Sociology		✓		✓						
Boutyline, Arseniev-Koehler & Cornell (2023)	Sociology	✓	✓		✓						
Hu et al. (2020)	Urban. Eng					✓			✓		
Murray et al. (2020)	Urban. Eng			✓		✓					
Kim et al. (2021)	Urban. Eng										
Niu & Silva (2021)	Urban. Eng			✓	✓	✓					
Johnson et al. (2023)	Psychology	✓	✓								
Durrheim et al. (2023)	Soc. Psych.	✓	✓		✓						
Garg et al. (2018)	General		✓		✓						
Grand et al. (2022)	General	✓	✓		✓		✓				
Charlesworth, Caliskan & Banaji (2022)	General	✓	✓		✓						
Lewis et al. (2023)	General		✓			✓					
Aceves & Evans (2024)	General		✓								
Kim, Askin & Evans (2024)	General		✓			✓					

Through this meticulous and methodical approach, we aimed to ensure a comprehensive and unbiased coverage of the literature. This strategy was crucial in facilitating a robust analysis of the current state of word embedding applications in social science research, thereby contributing significantly to our understanding of the field. This survey focuses the foundational role of the word2vec model introduced by Mikolov et al. (2013a) and we review the literature from 2013 to the present (2024). In addition to the word2vec application, we also acknowledges earlier research, including the seminal work of Rubenstein & Goodenough (1965) from 1965.

Word embedding models

We briefly review the technical aspects of word embedding models before diving into the main part of our survey. Since our focus is the literature that uses word2vec models, this section discusses the word2vec algorithm. This section ensures that the presented article provides sufficient technical prerequisites of word2vec, while in-depth explanations are relegated to relevant literature sources.2

Word embedding models learn the relationships between words in sentences in the data and return embedding vectors based on their learned model. The embedding vectors are considered to represent the semantics of the words in the sentence, and the distributional hypothesis Harris (1954) supports this assumption. The theory asserts that a word is determined by the surrounding words, and experiments with human subjects have validated that word embedding models can capture semantics similar to human cognition (Caliskan, Bryson & Narayanan, 2017). This prompts social scientists to assume that studying word embedding models learned from the text of interest can reveal how humans process information embedded in the text.

Evaluations of word embedding models roughly take two forms: intrinsic evaluation and extrinsic evaluation. Typical tasks for intrinsic evaluations are analogy and similarity. Many datasets for intrinsic evaluation are publicly available and accessible (Analogy (Miller & Charles, 1991; Charles, 2000; Agirre et al., 2009; Rubenstein & Goodenough, 1965; Bruni, Tran & Baroni, 2014; Radinsky et al., 2011; Huang et al., 2012; Luong, Socher & Manning, 2013; Hill, Reichart & Korhonen, 2015), Similarity (Mikolov et al., 2013a; Mikolov, Yih & Zweig, 2013)). For extrinsic evaluation, they often use word embedding models for translation or sentiment analysis. See Wang et al. (2019) for recent evaluation methods.

Although there are various word embedding models, social scientists have favored word2vec as the most popular model due to its simplicity, high user-friendliness, and longstanding usage (Chaubard et al., 2019; Rong, 2014). Thus, this survey article primarily focuses on the application of word2vec in social science research while also covering other language models that describe the new trend of this interdisciplinary research.

word2vec: a popular word embedding model

We begin the explanation of the technical background of word2vec by clarifying the terminology of “word2vec,” because the term does not designate a specific neural network model or algorithm but rather refers to a software program and tool. Pedagogically speaking, word2vec denotes software that combines a learning algorithm and a training method (Chaubard et al., 2019). For such packages, the continuous bag of words model (CBOW) and the skip-gram model are common implementations within word2vec, and the negative sampling algorithm is a widely-used technique (Mikolov et al., 2013a, 2013b; Goldberg & Levy, 2014; Mikolov, Yih & Zweig, 2013; Mikolov et al., 2015). While computer science articles usually describe the details of their models, social science articles often use word2vec as if it refers to a specific neural network model, and discussions of learning algorithms and training methods are not very common in social science articles except in those discussing research methods such as econometrics or statistical models. Therefore, when a article mentions only “word2vec” for their methods, it may use the CBOW or the skip-gram model3 .

CBOW, skip-gram model and SGNS model

The CBOW and skip-gram solve a “fake” problem that predicts what words appear in a given text. The main goal of the model is not to make predictions but to acquire the parameters obtained by solving the prediction problem as a low-dimensional representation of words. In other words, the prediction problem solved by word embedding is only to learn the parameters, and the prediction problem itself is a problem that is not of central interest. Given this background, we refer to prediction problems that word embedding solves to learn the low-dimensional representation as “fake” problems. To explain this fake problem in the word embedding model, we often use the terms “word” and “context” 4 . Target is the element in the text and is often a word. Context is the set of elements around the target and is often the set of words.

Consider the sentences, The stick to keep the bad away. The rope used to bring the good toward us. When the target is rope, its context could be [bad, away, the, used, to, bring] with a window size of three. The window size determines the context size and the number of elements to capture from the target. This example considers the three words to the left and right of the target word. While both the CBOW and skip-gram models solve the prediction model using the context and target, they differ in which part predicts the other. The CBOW model predicts the target w using the context c, P(w∣c). Conversely, the skip-gram model predicts the context using the target, P(c∣w). The most used word2vec is the skip-gram with negative sampling (SGNS) model that combines the skip-gram model and negative sampling algorithm because this combination is provided in the popular ready-to-use library gensim (Rehurek & Sojka, 2010a, 2010b).

The SGNS model considers the distribution, p(d∣w,c), where d takes 1 when a pair of target w and context c is observed in the data, 0 otherwise. The SGNS model maximizes the following conditional log-likelihood (Dyer, 2014),

(1) ∑(w,c)∈D{logp(d=1∣c,w)+kEw¯∼qlog⁡p(d=0∣c,w¯)}

where q is the noise distribution in negative sampling, and k is the sample size from the noise5 ; D is the set of paris of target w, context c, Ew¯∼q represents the expectation under the probability distribution of q(w¯), and k refers to the number of negative samples6 .

The SGNS model especially calculates the conditional probability p(d=1∣c,w) by σ(vc⋅vw), where σ(x) is sigmoid function and vw,vc∈Rd. In other words, it seeks the parameter vw,vc that maximizes the above conditional probability, which are the embedding vectors of interest.

The word2vecf model can construct an arbitrary context for the word2vec model. While a typical word2vec model constructs a set of words around the target word in its learning process, the word2vec model does not necessarily use the surrounding words of the target word for a set; it can be arbitrary. Since context determines the relationship between targets and words in the data, we can define the context based on external information. Levy & Goldberg (2014), for example, propose the word2vecf model, a modified version of the word2vec model, which obtains word embedding vectors by constructing the context based on the dependency between words. Moreover, the relationships to be learned can be in non-text data as we discuss in “Non-text”.

Taxonomy of applied methods and labeling literature

Now, we turn to the main focus of this survey article, which provides an overview of a new emerging trend that applies word embedding models to social science research. To give a clear overview of the applications of word embeddings in interdisciplinary research, we first develop a taxonomy of the analytical methods used in the surveyed social science articles. We then label the literature based on the developed taxonomy. This approach allows us to review the studies from a concrete perspective on the trend of word embedding applications, which spans a wide range of disciplines.

We first review the literature on applications of word embedding models for social science. Our taxonomy introduces the labels of popular methods found in the surveyed literature. This method-oriented classification benefits us when surveying interdisciplinary topics. The nature of these topics prevents us from discussing each article according to its research topics or questions because showcasing such highly diverse outcomes dilutes the visibility of the survey. Therefore, instead of following the literature of each research field, we study the way the surveyed articles use word embedding models for their analyses.

We discuss our taxonomy and labels with the literature in “Taxonomy”, and we summarize the labels in Table 1. We define eight labels and summarize the descriptive definitions of the analytical methods. We also list the surveyed articles in Tables 2 and 3 with the corresponding labels. This process identifies several important research orientations in the literature. For example, we identify a popular method that constructs variables for study using reference words (W/reference label). In this survey, we also observe a new line of literature that applies word2vec to non-textual data, such as relationships between users on web platforms.

Labeling literature

The central aim of this survey is to label the literature on applications of word embedding models for social science. To this aim, we first build the taxonomy on the literature in “Taxonomy” and label them based on them.

In addition, we group the articles by their research topics based on the journals or conferences in which they were published. For working articles, we infer the topic based on the author’s faculty affiliation and past publication history to determine the research area, resulting in 14 research areas. Note that we understand that defining or classifying research fields is not a trivial task.

Taxonomy

This section describes the taxonomy built with the labels introduced in “Labeling Literature”. We discuss several representative articles for each methodological label.

Pre-trained models

Word embedding models extract information from text data for research questions. Obtaining low-dimensional representations from data allows us to extract the semantics or hidden patterns in the data. When social scientists are interested in a specific subject, they train a word embedding model on their own data, such as descriptions of central banks (Matsui, Ren & Ferrara, 2021; Baumgärtner & Zahner, 2021), judges (Ash, Chen & Ornaghi, 2020), organizations (Chen & Zhang, 2023), job openings (Khaouja et al., 2019), or smartphone applications (Kawaguchi, Kuroda & Sato, 2021). To study these topics, researchers build their own datasets and train their word embedding models on the data.

While most studies construct their own word embedding models, pre-trained word embedding models are also popular, with articles using pre-trained models published in prestigious journals (Bhatia, 2019; Kozlowski, Taddy & Evans, 2019; Garg et al., 2018). Pre-trained word embedding models are ready-to-use models that have already been trained on large corpora. Most pre-trained models are trained on general corpora and are open to the public, such as the Google News pre-trained model (Google, 2013) and pre-trained models on historical corpora in several languages (Hamilton, Leskovec & Jurafsky, 2016b). Models trained on general corpora are supposed to represent general semantics, allowing studies with these pre-trained models to answer broad research questions, such as evaluating biases in publications (Garg et al., 2018; Caliskan, Bryson & Narayanan, 2017) and culture (Kozlowski, Taddy & Evans, 2019).

Studies have demonstrated that employing human subjects to validate the findings of pre-trained models can aid our understanding of crucial human perceptions (Caliskan, Bryson & Narayanan, 2017; Bhatia et al., 2021; Zheng, 2020). Such studies often use models pre-trained on general corpora to understand common human biases. For example, management scientists combine surveys from human subjects and pre-trained models to investigate essential concepts in their discipline, such as perceptions of risk (Bhatia, 2019) and leadership (Bhatia et al., 2021). Additionally, because of their generality, pre-trained models help researchers validate theories proposed in their disciplines. For example, researchers in neuroscience propose features for interpreting word embeddings based on a method called neurosemantic decoding (Chersoni et al., 2021).

Overfitting models

Rather than focusing on general social phenomena, social science research often targets specific subjects or classes. In such studies, researchers often “overfit” word embedding models on corpora that represent specific subjects. Overfitting is typically avoided by computer scientists to ensure the generality of their trained models since overfitted models do not perform well on computational tasks such as analogical reasoning or generating language models, as discussed in “Word Embedding Models”.

However, social science researchers are also interested in the results from overfitted models because these models can capture the specificity of the data. There is a line of research, for example, that investigates human biases, such as stereotypes, “embedded” in the text of interest (Bhatia & Bhatia, 2021; Ash, Chen & Ornaghi, 2020; Kroon, Trilling & Raats, 2021; Garg et al., 2018; Charlesworth, Caliskan & Banaji, 2022; Boutyline, Arseniev-Koehler & Cornell, 2023; Durrheim et al., 2023). Word embedding models learned from the corpora can reveal biases where certain words are overfitted with specific human characteristics, such as gender, and these computed biases are similar to the biases humans hold (Caliskan, Bryson & Narayanan, 2017). In the context of fairness in machine learning, significant computer science literature proposes methods for debiasing such biases (Bolukbasi et al., 2016; Zhao et al., 2018; Brunet et al., 2019; Font & Costa-Jussa, 2019) and evaluates their validity (Gonen & Goldberg, 2019).

Working variable

With trained word embedding models, most surveyed articles construct variables that embody the concepts or research questions to be examined in their study, while some articles use word embedding vectors for clustering (Hu et al., 2020; Nanni & Fallin, 2021; Zheng, 2020; Khaouja et al., 2019; Kim et al., 2021) or prediction (Bhatia et al., 2021; Richie et al., 2019; Lee & Kim, 2021; Wang, 2019) tasks. To clarify this trend, we call this procedure the working variable” using an analogy from working hypothesis,” which is a well-used term in social science research (Taylor, 2021). A working hypothesis is testable using empirical analysis and is often transformed from a conceptual hypothesis or theoretical hypothesis. While a theoretical hypothesis is a conceptual description of the hypothesis to be proved in the research, a working hypothesis describes an executable procedure or test that can prove the theoretical hypothesis. This method allows social scientists to transform a theoretical hypothesis into a working hypothesis that can be tested by experimental or observational research.

In this survey, we define “working variable” as a variable that is a proxy variable of theoretical interest. With working variables, researchers investigate theoretical hypotheses in various fields and external objects such as human perception. For example, to analyze stereotypes in the documents of interest, researchers can use word embedding to calculate some working variables that proxy stereotypes (Garg et al., 2018). Not only human perception, but also qualitative concepts can be defined as working variables such as the speed of semantics in documents like research articles (Soni, Lerman & Eisenstein, 2021; Toubia, Berger & Eliashberg, 2021), law documents (Soni, Lerman & Eisenstein, 2021), and the plots of movies and TV shows (Soni, Lerman & Eisenstein, 2021; Toubia, Berger & Eliashberg, 2021).

Defining working variables allows us to analyze social phenomena or concepts of interest through concepts or theories in social science. For example, working variables can be proxies of important concepts, such as opacity (Boyce-Jacino & DeDeo, 2020) or grammar (Čech et al., 2019). With embedding models, authors can construct their proxies for their research question (Johnson et al., 2023). Such variables are often defined as a scalar value obtained by the embedding model learned in the data of interest. Calculating these working variables usually employs the reference words described in “Reference Words”. Defining not only a single working variable but also multiple working variables clarifies the research questions. Toubia, Berger & Eliashberg (2021), for instance, quantified the shape of stories by defining several variables such as speed, volume, and circuitousness in the text.

There are studies that directly calculate vectors obtained from word embedding models and some researchers use word embedding models to estimate their own statistical model for causal inference (Wang, 2019) and equilibrium estimation (Kawaguchi, Kuroda & Sato, 2021), rather than analyzing the subject represented using the embedding vectors.

Reference words

To construct working variables, the literature often introduces “reference words” that serve as a point from which researchers calculate the distance of word embedding vectors. In this method, “reference” words represent a concept to be analyzed.

For example, Garg et al. (2018) analyze the gender stereotypes in text over one century by calculating the relative similarity between the words related to gender (men or women) and the words related to specific occupations. To calculate the relative similarity, they first calculated the distance between the word embedding vectors of words related to men and the words describing specific occupations. They also calculated the same distance for women-related words. Then, they computed the relative differences between these two distances. When the relative distance with an occupation is large, it means that the stereotype in the text is considerable. For example, they determined that “engineer” is close to the men-related words in the historical corpus and argued that this suggests the existence of stereotypes in the corpus. The article showed that the measured gender stereotypes are correlated with occupational employment rates by gender. They also conducted the same analysis for ethnic stereotypes.

There are two advantages to using reference words for analysis. The first is that introducing references improves the interpretability of the results; if we use the Subjectivity Lexicon as the reference word, we can measure the degree to which the word of interest is subjective (Matsui, Ren & Ferrara, 2021). Secondly, this method is free from the problems associated with coordinate systems when comparing different models detailed in this point in “Comparing the same words”.

Comparing the same words

Some prefer simple methods of analyzing the similarity between words. For example, this category contains the analysis of enumerating words that are similar to a certain word, which is often done in word embedding tutorials because the similarity of each word is calculated to list similar words. Some articles have used this simple analysis as seminal work and have yielded useful findings (Chanen, 2016).

Calculating the similarity between the same words from different models could also answer important research questions for social science. The same word often has different meanings depending on the speaker and the time period in which the word was used. Examining such diachronic changes in semantics or meaning by the speaker can address profound research questions in social science. Nyarko & Sanga (2020), for example, examined the difference in perception between experts and laypeople by investigating whether these two groups use the same word to describe different semantics according to their profession.

When comparing the semantic differences of the same word from two different documents, we need to train two separate word embedding models on each document. In such a situation, we should note that we cannot directly compare the two models, even if they were trained with the same algorithm because algorithms typically use an arbitrary coordinate system with each training. To address this issue, Hamilton, Leskovec & Jurafsky (2016a) uses orthogonal Procrustes to align different models to reveal historical changes in the semantics of words, and some other studies follow this method (Nyarko & Sanga, 2020). Orthogonal Procrustes, however, may not be stable (Gillani & Levy, 2019). Therefore, an improved method using the dynamic word embedding model, which explicitly incorporates semantic changes, has been proposed (Gillani & Levy, 2019; Yao et al., 2018), and some articles use the dynamic word embedding model to analyze semantic changes (Matsui, Ren & Ferrara, 2021). Thanks to recent studies on the statistical properties of word embedding models (Arora et al., 2016), researchers have proposed a way to compare different models (Zhou, Huang & Zheng, 2020) without a model alignment, which will advance the use of word embedding models from a more social science perspective. Additionally, this problem can be avoided if the comparison is done within the same model and is projected onto a scalar value. Garg et al. (2018) calculate stereotypes as scalar values and investigate the trajectory of the values over time. As discussed in “Reference Words”, this is the second advantage of introducing reference words that allow us to avoid the problems stemming from the issue of word embedding models using an arbitrary coordinate system in their learning.

Non-text

This survey observes a recent trend that applies word2vec to non-textual data. As discussed in “CBOW, Skip-gram Model and SGNS Model”, word embedding can learn relationships between not only words but also any entities such as metadata (Rheault & Cochrane, 2020) and symbols (Tshitoyan et al., 2019), in addition to text data. This subsection also surveys works that apply word2vec algorithms (skip-gram or CBOW) to digital platforms and geographic data (Mikolov et al., 2013a, 2013b; Goldberg & Levy, 2014; Mikolov, Yih & Zweig, 2013; Mikolov et al., 2015).

The relationships that word embedding models learn are not limited to words in text. Word embedding models can obtain low-dimensional representations from relationships composed of non-text data. For example, Hu et al. (2020) and Niu & Silva (2021) obtain embedding vectors of Points of Interest (POI) from geological data describing the relationships between POIs. With their clustering algorithm, they suggested it can discover the usage pattern of urban space with reasonable accuracy. Murray et al. (2020) noted an important theoretical insight for analyzing geographic data with word2vec. They revealed that word2vec is mathematically equivalent to the gravity model Zipf (1946), which is often used to analyze human mobility, and conducted experiments with real datasets to validate their findings.

Word embedding models can also learn relationships between chemical equations and the text around them in a research article. Tshitoyan et al. (2019) apply word2vec to academic articles in chemistry to learn the relationship between text and symbols (chemical formulae). They demonstrate that potential knowledge about future discoveries can be obtained from past publications by showing that the obtained embedding model can recommend discoveries several years in advance of the actual discovery. Not only symbols but also other information can be embedded. Rheault & Cochrane (2020) analyze politicians’ speeches using the word2vec algorithm to embed information about politicians’ ideologies along with the text data of their speeches.

Non-text data may yield useful insights even when text data is available in the dataset. Kim, Askin & Evans (2024) learn the representation of music track by applying the word2vec algorithm to the sequence of music listening history. Waller & Anderson (2021) adopt a modified version of the word2vec algorithm, using algorithm called word2vecf, which is a modified version of word2vec (Levy & Goldberg, 2014; Goldberg, 2017). to analyze the posting behavior of Reddit users. They obtained embedding vectors by learning the relationships between Reddit users and communities (subreddits). It is worth noting that they do not analyze the textual data of Reddit posts to investigate users’ behavior. Studying Reddit text data might unveil the attributes of Reddit users, such as demographic information, preferences, and age groups. However, user comments do not always reflect such meta-information. Some supervised machine learning models may predict such meta-information; there are successful unsupervised methods that predict the characteristics of text, such as sentiment analysis (Pennebaker, Francis & Booth, 2001; Bird, Klein & Loper, 2009; Hutto & Gilbert, 2014) or moral foundations dictionary (Graham et al., 2011; Haidt & Graham, 2007). Moreover, the validity of training data is difficult to prove due to issues regarding the annotator’s subjectivity. To overcome this problem, the authors used a data-driven and linguistically independent method that characterizes subreddits only from the perspective of “what kind of users are posting in the community.” By setting reference “words” (subreddits), they created indicators such as age, gender, and partisanship. For example, to calculate the working variable of “partisanship,” they chose opposing communities in Reddit: “democrats” and “conservatives.” Then, they calculated the relative distance for each community. Applying our taxonomy to Waller & Anderson (2021) reveals that, while they do not use text, they share characteristics with other studies in their methodologies. They define a working variable (partisanship) using reference words (subreddits). In addition, they are similar to the analysis by Toubia, Berger & Eliashberg (2021) in terms of setting multiple working variables.

Several studies apply the word embedding algorithm to learn relationships between text and non-text data or between non-text entities. As discussed in the introduction, this survey primarily focuses on the applications of word embedding models and algorithms. Therefore, the non-text label does not cover embedding models for non-text data if they are not word embedding models, such as graph embedding (Goyal & Ferrara, 2018; Barros et al., 2021), image embedding (Liu et al., 2020; Baltrušaitis, Ahuja & Morency, 2018), or network embedding (Cui et al., 2018; Zhou et al., 2022a).

Cosine similarity or euclidean distance?

We find that many articles examine the similarity between embedding vectors when they employ the methods discussed so far such as reference words (“Reference Words”) or non-text (“Non-text”). For similarity measurement, most use a cosine similarity but Euclidean distance ( L2 norm) is also popular. Because the similarity between vectors can be interpreted as the distance between vectors, we have various options to define the distance between vectors7 . However, it is not trivial whether the results can change depending on the measure we use for analysis. Some articles analyze the robustness of the results using multiple options, such as Euclidean distance and cosine similarity Garg et al. (2018).8

Toubia, Berger & Eliashberg (2021), in their supplementary information, argues that Euclidean distance is richer than cosine similarity that only measures the angle. Indeed, cosine similarity is not a perfect alternative to Euclidean distance. Cosine similarity can disregard the information on the distance between two given vectors. Figure 1 depicts five different vectors. Vectors 1 and 2 have the same angle θ1, and therefore, the cosine similarity of these is one. In contrast, the Euclidean distance between the two is d, which means they are different in terms of Euclidean distance.

Figure 1 Empirical relationships between cosine-similarity vs. euclidean distance.

Note: Schematic illustration of three different vectors. Vector 1 and Vector 2 have the same angle θ1, but Vector 1 is longer than Vector 2. The angle between Vector 2 and Vector 3 is θ2 but the distance between Vector 2 and Vector 3 is the same as the distance between Vector 1 and Vector 2. The distance between Vector 4 and Vector 5 is close, but it has a certain angle.

It is important to recognize that equal Euclidean distances do not always correspond to equivalent cosine similarity values and that Euclidean distance cannot always be interchanged with cosine similarity, nor vice versa. The Euclidean distance between Vectors 2 and 3 is the same as the Euclidean distance between Vectors 1 and 2, but these two pairs have different angles ( θ1 and θ2). Figure 1 also depicts that using the normalized Euclidean distance eliminates the distance between Vectors 1 and 2. The normalized Vector 1 will be on the same point as Vector 2, which is on the unit circle. In this case, these two vectors are the same from the perspective of both cosine similarity and Euclidean distance. Also, we should note that small distances do not always mean small cosine distances. For example, Fig. 1 depicts Vectors 4 and 5 is small in Euclidean distance compared to the other vectors, but their angle is θ1 (approximately π/4); therefore, the cosine similarity between them is not small.

Specifically, the relationship between two measurements becomes dismissed when θ is small or around π. Let us consider the distance between two points in a unit circle: A (0,1) and B (cos⁡θ,sin⁡θ).We can consider two points in a unit circle by (x,y) and (xcos⁡θ−ysin⁡θ,xsin⁡θ+ycos⁡θ), where the angle between the two is θ. Since we only study the distance between the two, we set x=1 and y=0 without loss of generality. The distance between two is (2(1−cos⁡θ))1/2, and the cosine similarity is cos⁡θ. Therefore, the error between the Euclidean distance of normalized vectors and cosine similarity is a function of θ, and that is

(2) e(θ)=cos⁡θ−(2(1−cos⁡θ))1/2.

Since the first derivative of e(θ) with respect to θ is

(3) e′(θ)=−sin⁡(θ)(1+(2(1−cos⁡(θ))1/2)),

the relationship between the two metrics become zero around ( θ=π or 0) where angle θ∈R. This fact implies that the two metrics are incompatible when their angle is small or large. We also note that the fact that the angle is not always relevant to the distance.

To further discuss the comparison between the cosine similarity and Euclidean distance, we study the empirical relationship between them using the Google News pre-trained word2vec model that contains the embedding vectors of 3,000,000 words Google (2013). We construct 1,500,000 random word pairs and plot the relationships in Figs. 2A and 2B. While Fig. 2A demonstrates that the two metrics are correlated, their correlation is not strong (the Pearson correlation coefficient ρ=−0.357), which implies that using cosine similarity would lose some information and vice versa. Figure 2A also demonstrates that the cosine similarity distributes uniformly where the Euclidean distance is small, meaning that the two different pairs of vectors with the same Euclidean distance can assume different values in their cosine similarities. This deviation becomes large where the Euclidean distance is small. This finding is consistent even when we normalize the metrics. Figure 2B demonstrates the empirical relationship between the Euclidean distance of normalized vectors and cosine similarity. While they capture the difference of two word embedding vectors in a similar way (the Pearson correlation coefficient ρ=−0.991), they are not the same. The Euclidean distance of normalized vectors is more sensitive than cosine similarity in which the distance between two points is close (i.e., θ is small) or large (i.e., θ is around π).

Figure 2 Schematic comparison: cosine-similarity vs. Euclidean distance.

Note: We built 1,500,000 pairs of two words by randomly picking words from Google News pre-trained word2vec model Google (2013). Then, we plot the cosine similarity and Euclidean distance between the pairs and the relationship between the cosine similarity and normalized Euclidean distance. We normalize word embedding vectors such that the norm of each embedding vector is 1.

The above discussion suggests that the cosine similarity and Euclidean distance can qualitatively return in a similar manner, but they are not always compatible. Notably, the cosine similarity can capture the difference between two vectors when their Euclidean distance is small. However, when we study the normalized vectors, the Euclidean distance captures the differences, but the cosine similarity does not when the Euclidean distance is small. Given this, we should know what characteristics we intend to capture when selecting a metric.

These findings also imply that the Euclidean distance and cosine similarity (angle) capture different characteristics. Recently, some studies in computer science literature have proposed word embedding models that explicitly model this relation, such as hierarchical relationships (Tifrea, Bécigneul & Ganea, 2018, Nickel & Kiela, 2017; Vendrov et al., 2016; Vilnis & McCallum, 2015). For example, (Iwamoto, Kohita & Wachi, 2021) proposed the embedding models that use the polar coordinate system and illustrate that their model captures the hierarchy of words in which the radius (distance) represents generality, and the angles represent similarity.

Pitfalls with cosine similarity for similarity measurement with word embedding vectors

Among the surveyed articles, cosine similarity is the most popular similarity measurement but the recent literature points out several critical pitfalls of analysis with this popular measurement. Zhou et al. (2022b) demonstrate that the cosine similarity measurement can underestimate the similarity between word embedding vectors compared to human judgments. They attribute this underestimation to differences in word frequency, suggesting that low-frequency and high-frequency words have different geometric representations in the embedding space. Steck, Ekanadham & Kallus (2024) also detects issues with cosine similarity measurement, showing that it can return arbitrary similarity values in certain settings using regularized linear models. Wannasuphoprasit, Zhou & Bollegala (2023) identify a problem where cosine similarity underestimates the actual similarity between high-frequency words as Zhou et al. (2022b). They propose a method to mitigate this problem that discounts the ℓ2 norm of a word embedding by its frequency to correct this underestimation, demonstrating that the proposed method effectively resolves the issue.

Conclusion

This survey comprehensively overviewed the emerging interdisciplinary research trend. We survey the literature on the application of word embedding models to social science. Since this research trend covers a wide range of research areas, we first construct the taxonomy of the method used in the surveyed articles, constructing the labels of popular methods. We then discuss the literature with the constructed labels and this helps us with documenting and bridging the literature in different research areas from neuroscience to humanity. In addition, we discuss several important issues regarding the similarity measurement, which researcher should note in their application.

We have made several contributions to the literature and the science community. First, this survey detects the emerging trend of an application in social science that uses word embedding algorithms, and we document this trend not only by focusing on a single research area or discipline. This survey would serve “a hub” for researchers from various academic fields to refer to findings from other disciplines when using word embedding models for social science analysis. The second contribution is the introduction of the taxonomy of the methods used in the surveyed article. Based on this, we provide the labels that would help researchers classify their work based on their methods. Before this work, the literature has not particularly contextualized their work based on the way they use word embedding because of the lack of notions that bridge the literature. The present research also provides researchers with useful guidance and a well-organized framework for how to explore or classify these applications of word embedding models in social sciences. Lastly, we contribute to the literature by summarizing several important issues when applying word embedding to social science research. More specifically, the survey discusses some pitfalls of using popular similarity measurements. We would like to expect that this survey article harnesses the researcher working on this interdisciplinary area to explore and contextualize their studies, connecting their work across different research areas.

While the present article contributes to the science community in the aforementioned points, it has limitations as the other works do. First, we mainly focused on the application of the word2vec algorithm and did not deeply explore ones utilizing other embedding algorithms or neural-based approaches. For example, this survey did not cover the sentiment analysis or topic modeling by machine learning including the embedding-based method such as BART (Lewis et al., 2020). In addition, the presented work did not deeply discuss the methodological discussion about word embedding. There are a substantial amount of topics about the theoretical aspect of word embedding algorithms as the famous quote goes “Good question. We don’t really know.” in Goldberg & Levy (2014). The further direction for this literature could establish the theoretical foundation and empirical evidence that provides a framework or procedure for researchers who apply word embedding algorithms to social science research.

Additional Information and Declarations

Competing Interests

Author Contributions

Data Availability

1 Portions of this text were previously published as part of a preprint (Matsui & Ferrara, 2022) and the author’s dissertation (Matsui, 2022)

2 Kozlowski, Taddy & Evans (2019) summarize the history of word embedding models for social scientists, and Chaubard et al. (2019) provide an excellent description of word2vec.

3 In this survey, we review the articles (Ash, Chen & Ornaghi, 2020; Lee & Kim, 2021; Caliskan, Bryson & Narayanan, 2017) that uses GloV model (Pennington, Socher & Manning, 2014; Kawaguchi, Kuroda & Sato, 2021) that uses fastText model (Bojanowski et al., 2017).

4 While “target” is also popularly used instead of “word,” we stick to using “word” in this survey to avoid potential confusions.

5 In the literature, it is often mentioned that this log-likelihood maximization with the noise distribution is considered as Noise Contrastive Estimation (NCE) (Gutmann & Hyvärinen, 2010), and the SGNS model is one of the variations of NCE.

6 Gutmann & Hyvärinen (2010) describes the details, and this article also follows the notation and equations of that article.

7 We also should note that cosine similarity is not a distance metric.

8 Regarding the robustness check, Ash, Chen & Ornaghi (2020) also studied the correlations between their stereotype measurements of 100 and 300 dimension embedding vectors. In addition, they tested three sets of window sizes in their robustness check.

The authors declare that they have no competing interests.

Akira Matsui conceived and designed the experiments, performed the experiments, analyzed the data, performed the computation work, prepared figures and/or tables, authored or reviewed drafts of the article, and approved the final draft.

Emilio Ferrara conceived and designed the experiments, prepared figures and/or tables, authored or reviewed drafts of the article, and approved the final draft.

The following information was supplied regarding data availability:

This is a literature review.

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
