# Peer review of "Word embedding for social sciences: an interdisciplinary survey"

_PeerJ Computer Science, doi:10.7717/peerj-cs.2562_

## Round 0.1 · original submission · Major Revisions

Dear authors,

Thank you for submitting your article. Reviewers have now commented on your article. Your article has not been recommended for publication in its current form. However, we encourage you to address the concerns and criticisms of the reviewers, particularly in terms of readability, validity of the findings, and overall quality, and to resubmit your article once you have updated it accordingly.

Reviewer 1 has asked you to provide specific references. You are welcome to add them if you think they are relevant. However, you are not obliged to include these citations, and if you do not, it will not affect my decision.

Best wishes,

·

Basic reporting

This paper comprehensively surveys the burgeoning application of word embedding techniques across the social sciences. Through a systematic search of journals, proceedings and preprint servers, the authors catalog over 100 studies utilizing word2vec and related models to derive low-dimensional representations of text for analyses. A taxonomy is proposed categorizing applications by research topic and analytical method, with the latter classified as employing pre-trained or overfit models, defining variables from embeddings, comparing references, assessing similarity metrics, evaluating cross-model differences and applying the approach to non-text data. While providing a useful organizing framework, deeper scrutiny of individual works and quantitative validation of the taxonomy are needed to substantiate claims of a holistic view. Nevertheless, this large-scale review serves to integrate knowledge across disciplines and illuminate promising directions like advancing non-textual applications to stimulate further methodological and theoretical progress in this interdisciplinary field.

Experimental design

See additional comments.

Validity of the findings

See additional comments.

Additional comments

1.The literature review could be more comprehensive by including relevant works from computer science conferences to provide a fuller picture of word embedding applications.

2.Definitions of some technical terms like "target" and "context" lack adequate rigor for specialist readers.

3.Taxonomy construction warrants a more detailed methodology section to ensure classifications are reliable and validate hierarchy.

4.Lack of quantitative metrics to systematically gauge studies against taxonomy prevents objective assessment.

5.Subjective choices in search strategy may introduce biases; a reproducibility appendix clarifying protocols would strengthen findings.

6.Lack of conceptual framework linking existing knowledge hampers identification of theoretical gaps for future research.

7.Conclusions do not speculate on broader impacts or outline implications for academic practice, limiting work's influence.

8.Readability and language quality could be improved in some sections to enhance comprehension for a global readership.

9.The cited literature is not new enough. For example, you can refer to some new integrated social media analysis methods:

(1)Gong, P., Liu, J., Zhang, X., & Li, X. (2023, June). A Multi-Stage Hierarchical Relational Graph Neural Network for Multimodal Sentiment Analysis. In ICASSP 2023-2023 IEEE International Conference on Acoustics, Speech and Signal Processing (ICASSP) (pp. 1-5). IEEE.
(2)Xiao, L., Wu, X., Yang, S., Xu, J., Zhou, J., & He, L. (2023). Cross-modal fine-grained alignment and fusion network for multimodal aspect-based sentiment analysis. Information Processing & Management, 60(6), 103508.
(3)Qian, F., Han, J., He, Y., Zheng, T., & Zheng, G. (2023, July). Sentiment Knowledge Enhanced Self-supervised Learning for Multimodal Sentiment Analysis. In Findings of the Association for Computational Linguistics: ACL 2023 (pp. 12966-12978).

Cite this review as

·

Basic reporting

no comment

Experimental design

no comment

Validity of the findings

no comment

Additional comments

This paper introduces the development of machine learning models by computer scientists to extract key information from complex data, where these models learn low-dimensional representation patterns. This advancement not only benefits computer scientists but also proves advantageous for social scientists since human behavior and social phenomena are embedded in complex data. To document this emerging trend, the authors conducted a survey of recent studies applying word embedding techniques to human behavior mining. They established a taxonomy to illustrate the methods and procedures used in the surveyed papers, highlighting the latest trends in applying word embedding models to non-textual human behavior data.

However, the following suggestions should be taken into consideration.

1. Clear Identification of Contributions: It is recommended that the authors clearly identify the main contributions of the paper in the abstract. This will help readers understand the uniqueness of the research and why it is significant in the relevant field.

2. Creation of a Core Framework Diagram: To better visualize the core concepts and methods of the paper, it is suggested that the authors include a core framework diagram in the text. Such a visual aid can assist readers in quickly grasping the overall structure and flow of the research.

3. Experimental Validation and Comparison: Authors are advised to emphasize the importance of experimental validation in the abstract and clarify whether experiments were conducted. It is preferable to compare the proposed method's performance with the current state-of-the-art methods, highlighting the effectiveness and superiority of the research. If experiments have not been conducted, the plan for future experimental validation can be mentioned.

4. Clarity and Structure: While the abstract provides a clear description of the research content and structure, it is suggested to explicitly emphasize the uniqueness and innovation of the study to captivate the reader's interest.

5. Language Expression: Careful attention should be given to grammar and wording in the abstract to ensure clear and unambiguous expression throughout the paper.

---

## Round 0.2 · Major Revisions

Dear authors,

Thank you for submitting your revised article. The previous reviewers could not be contacted and we had to find a new reviewer. Based on this review, your article has not yet been recommended for publication in its current form. However, we encourage you to address the concerns and criticisms of the reviewer and to resubmit your article once you have updated it accordingly. This new Reviewer 3 has asked you to provide specific references. You are welcome to add them if you think they are relevant. However, you are not obliged to include these citations, and if you do not, it will not affect my decision.

Best wishes,

Reviewer 3 ·

Basic reporting

Outline

First of all, I would like to congratulate the author(s) for their work. Please be assured that I will evaluate the study only from a scientific point of view, without prejudice.

This work is valuable and prudent in the computer science community and social media area. Also, I can't end without saying that it is worth reading.

In the study, the author(s) survey paper aims to provide an entry point for all interested researchers and data mining experts to learn key algorithms, word embedding techniques, and growing innovations for application to human behavior mining.

Also, using the powerful lens of literature, the current study reviews the main approaches in the field, discusses trends in different eras, discusses the strengths and limitations of these methods, and summarizes the essential findings and future directions for research in this area.

Experimental design

Design and Writing

The article should be prepared by considering the journal template writing rules (see: https://peerj.com/about/author-instructions/cs).
Although the article was prepared with great effort, there are typographical errors in the most prominent parts of the article. These deficiencies can't be tolerated for the valuable study.
- The article should be completely revised from the use of abbreviations to reference notation.
ex1. This section describes the taxonomy built upon on the labels constructed in Section 3.1 and we discuss several representative papers for each methodological label.
- Also, choose correctly the verbs that emphasize the achievement results of the literature.
- Attention should be paid to the use of abbreviations.
- Some of your sentences should be shortened and clarified
- Incomplete and inconsistent sentences should be completed. In addition, the article should be cleared of all grammatical errors by using the grammar check tool (For example, Grammarly or Ginger).
- Note that mathematical expressions are defined in the text and each symbol is highlighted. (Page 10/19).
- I recommend you elaborate on the Conclusion section. Emphasize the contribution of this study to the literature and its importance for future studies.

Validity of the findings

In the abstract section, clearly emphasize your main motivation for the preparation of the article. The last sentence of the Abstract section must have been completed with a more striking sentence.

In the conclusion section, list your achievements with this study. Clearly express your contribution to future studies and literature.

Satisfactory technical information about the method used in the literature is not presented. No mathematical expressions, from the evaluation metrics used to the algorithms, are included in the article.
Please clearly describe the methods and algorithms analyzed.

Not only the methods and methods applied for human behavior mining but also those used in other SA studies should be investigated. Attention should be paid to machine learning deep learning, and optimization methods. Please check the additional comments section.

Provide the motivation for the conclusion part.

Additional comments

In addition, in order to examine the latest technology artificial intelligence and optimization algorithms, read the following articles and add all that seem relevant to your article.
2022, Deep-Cov19-Hate: A Textual-Based Novel Approach for Automatic Detection of Hate Speech in Online Social Networks throughout COVID-19 with Shallow and Deep Learning Models
2021, Performance Assessment of Artificial Intelligence-Based Algorithms for Hate Speech Detection in Online Social Networks https://doi.org/10.35234/fumbd.986500
2021, Metaheuristic Ant Lion and Moth Flame Optimization based Novel Approach for Automatic Detection of Hate Speech in Online Social Networks
2021, Sentiment Analysis in Social Networks Using Social Spider Optimization Algorithm
2019, Detection of Customer Satisfaction on Unbalanced and Multi-Class Data Using Machine Learning Algorithms
2018, Sentiment analysis using Konstanz Information Miner in social networks

Cite this review as

---

## Round 0.3 · Minor Revisions

Dear authors,

Thank you for submitting your revised Literature Review paper. Reviewers did not respond to the invitation for the revision. Your paper seems improved however it still needs minor revision and it will be better to address the following:

1. The Introduction section should contain a well-developed and supported argument that meets the goals set out.
2. How this review paper will contribute to the scientific body of knowledge should be clearly mentioned.
3. The coverage (both temporal and domain) of the literature and how the literature was distributed across time domains should be clearly provided.
4. Clearly reported, reproducible, and systematic methods should be provided in order to identify, select, and critically appraise relevant research.
5. Please provide a clearly defined research question for this literature review paper.
6. Equations should be used with equation number. Explanation of the equations should be checked. All variables should be written in italics. Definitions and boundaries of all variables should be provided. Necessary references should also be given.

Best wishes,

---

## Round 0.4 · accepted · Accept

Dear Authors,

I am grateful for your efforts in revising the paper. I am satisfied with the revised manuscript and believe it is now ready for publication.

Best wishes,